# Baseline Values of Circulating IL-6 and TGF-β Might Identify Patients with HNSCC Who Do Not Benefit from Nivolumab Treatment

**DOI:** 10.3390/cancers15215257

**Published:** 2023-11-02

**Authors:** Marco Carlo Merlano, Matteo Paccagnella, Nerina Denaro, Andrea Abbona, Danilo Galizia, Dario Sangiolo, Loretta Gammaitoni, Erika Fiorino, Silvia Minei, Paolo Bossi, Lisa Licitra, Ornella Garrone

**Affiliations:** 1Candiolo Cancer Institute, FPO-IRCCS Candiolo, 10060 Torino, Italy; mcmerlano@gmail.com (M.C.M.); danilo.galizia@ircc.it (D.G.); loretta.gammaitoni@ircc.it (L.G.); 2Translational Oncology ARCO Foundation, 12100 Cuneo, Italy; matteo.babeuf@gmail.com; 3Medical Oncology, Fondazione IRCCS Ca’ Granda Ospedale Maggiore Policlinico, 20122 Milano, Italy; nerina.denaro@policlinico.mi.it (N.D.); ornella.garrone@policlinico.mi.it (O.G.); 4Department of Oncology, University of Turin, 10060 Torino, Italy; dario.sangiolo@unito.it (D.S.); erika.fiorino@unito.it (E.F.); 5Post-Graduate School of Specialization Medical Oncology, University of Bari “A. Moro”, 70120 Bari, Italy; silvia.minei3@gmail.com; 6Medical Oncology, A.U.O. Consorziale Policlinico di Bari, 70120 Bari, Italy; 7Medical Oncology, Department of Medical and Surgical Specialties, Radiological Sciences, Public Health, University of Brescia, 25123 Brescia, Italy; paolo.bossi@hunimed.eu; 8Fondazione IRCCS Istituto Nazionale dei Tumori, University of Milan, 20133 Milan, Italy; lisa.licitra@gmail.com

**Keywords:** HNSCC, IL-6, TGF-β, nivolumab

## Abstract

**Simple Summary:**

Immunotherapy has positively modified the natural history of head and neck cancer (HNC), however, adequate biomarkers to identify resistant patients have not yet been identified. We analyzed 18 circulating Th1, antitumor, and Th2, protumor, cytokines in patients with relapsed/metastatic HNC treated with the immune checkpoint inhibitor nivolumab. Blood samples were collected at baseline (T0) and after 3 cycles of nivolumab (T1). Data extracted at T0 have been related to survival; the comparison between T0 and T1 explored the effect of immunotherapy. Elevated IL-6 and TGF-β values at T0 identified patients with poor survival. Increases of CCL-4, IL-15, IL-2 and CXCL-10, all Th1 cytokines, occurred between T0 and T1 but did not correlate with survival. We suggest that circulating cytokines may represent a tool to identify patients resistant to immunotherapy. The increase of circulating Th1 cytokines during treatment with nivolumab suggests that the drug modulates the tumor microenvironment. However, the lack of correlation between these changes and survival highlights that immunosuppressive mechanisms can prevail. This finding supports the combination of immune checkpoint inhibitors with agents that inhibit other immunosuppressive pathways.

**Abstract:**

Background: The immunotherapy of head and neck cancer induces a limited rate of long-term survivors at the cost of treating many patients exposed to toxicity without benefit, regardless of PD-L1 expression. The identification of better biomarkers is warranted. We analyzed a panel of cytokines, chemokines and growth factors, hereinafter all referred to as ‘cytokines’, as potential biomarkers in patients with head and neck cancer treated with nivolumab. Materials and methods: A total of 18 circulating cytokines were analyzed. Samples were gathered at baseline (T0) and after 3 courses of nivolumab (T1) in patients with relapsed/metastatic disease. The data extracted at T0 were linked to survival; the comparison of T0–T1 explored the effect of immunotherapy. Results: A total of 22 patients were accrued: 64% current heavy smokers, 36% female and 14% had PS = 2. At T0, ROC analysis showed that IL-6, IL-8, IL-10 and TGF-β were higher in patients with poor survival. Cox analysis demonstrated that only patients with the IL-6 and TGF-β discriminate had good or poor survival, respectively. Longitudinal increments of CCL-4, IL-15, IL-2 and CXCL-10 were observed in all patients during nivolumab treatment. Conclusion: In this small population with poor clinical characteristics, this study highlights the prognostic role of IL-6 and TGF-β. Nivolumab treatment is associated with a positive modulation of some Th1 cytokines, but it does not correlate with the outcome.

## 1. Introduction

Immunotherapy has achieved important improvements in the treatment of recurrent/metastatic head and neck squamous cell cancer (R/M-HNSCC), leading to a reproducible rate of long-term survivors both in front-line and in second-line treatment.

However, even in the most favorable population, more than 60% of treated patients die within 2 years from the beginning of treatment [1].

The tumor microenvironment (TME) dictates the way in which a tumor evades the immune response. Therefore, the immune signature of the TME may better identify patients who will benefit from immune checkpoint inhibitors (ICIs) [2].

It has long been known that cytokines can mirror the TME [3]. As a consequence, their study may represent an alternative to identify the patient’s TME and to follow its changes over time [4].

Cytokines produced by both immune cells and tumor cells in the TME may represent good candidate biomarkers as they are easy to obtain with peripheral blood, provide a non-invasive opportunity to monitor dynamic changes and are relatively cheap [4].

Several clinical studies have evaluated single cytokines as biomarkers of responses and survival rates in patients treated with immunotherapy, target therapy or chemotherapy [5,6,7,8].

However, the data are frequently conflicting and definitive results have not yet been reached [4].

Indeed, cytokines work as a network and the effect of a single cytokine is context-dependent. Therefore, the measurement of a panel of cytokines might be more appropriate to define a prognostic/predictive effect [4,9].

The aim of this study was to test a panel of circulating cytokines at two time points, T0 (baseline) and T1 (during treatment), to identify possible biomarkers in patients with head and neck cancer (HNC) treated with the ICI nivolumab. Furthermore, longitudinal changes possibly observed in the same cytokines could add information on the effect of nivolumab.

## 2. Materials and Methods

This paper reports an ancillary analysis of the Nivactor-GONO (Gruppo Oncologico Nord-Ovest) study (Eudract 2017-000562-30).

Study design is reported in Appendix A. Briefly, the Nivactor study is a single-arm, open-label, multicenter, phase IIIb real-life clinical trial. The study aimed to verify, in an unselected patient population with recurrent/metastatic HNC previously treated with chemotherapy, the toxicity of nivolumab-based immunotherapy. The analysis was conducted in patients accrued at two centers among those participants in the main trial: S. Croce and Carle General Hospital, Cuneo, Italy and Candiolo Cancer Institute, FPO-IRCCS, Candiolo (Turin, Italy).

Patients were retrospectively allocated into two groups using six-month overall survival (OS) as a cut-off point:

Group 1. Patients with OS below six months.

Group 2. Patients with OS above six months.

### 2.1. Plasma Collection

Blood samples were collected at baseline (T0) and after three shots of immunotherapy (day 28) (T1).

Twelve mL of peripheral blood samples in EDTA-treated Vacutainer (BD, Franklin Lakes, NJ, USA) were collected from each patient. Plasma samples were separated with centrifugation step at 800× *g* for 10 min at room temperature (RT) and immediately stored at −80 °C until use.

### 2.2. Cytokines

We selected 18 low molecular weight soluble proteins on the basis of their predominant Th1 (pro-inflammatory) or Th2 (anti-inflammatory) effects, their association with survival or their acknowledged immunosuppressive actions.

Interleukin (IL)-2, IL-12, IL-15, IL-21, C-X-C chemokine ligand (CXCL)-10 tumor necrosis factor (TNF)-α, C-C motif chemokine ligand (CCL)-2, CCL-4 and interferon (IFN)-γ were all considered Th1 cytokines. IL-4, IL-5, IL-10, IL-13 and CCL-22 were considered Th2 cytokines [10]. IL-6, IL-8 (associated with poor survival in many solid tumors) [11,12], transforming growth factor (TGF)-β and vascular endothelial growth factor (VEGF) were considered major immunosuppressive factors [13,14]. All the above interleukins, chemokines, interferons and growth factors are hereafter called as ‘cytokines’ for brevity.

### 2.3. Cytokine Measurement

Concentrations of all cytokines but IL-21 were measured using the Ella Simple Plex system (ProteinSimple™, San Jose, CA, USA) following the manufacturer’s instructions.

IL-21 was assessed using the ELISA method (R&D System^®^, Minneapolis, MN, USA).

In brief, as previously described [7], a two-fold dilution of every plasma sample was spun for 15 min at 1000× *g* and was inserted into a Simple Plex cartridge. The cartridge was then placed inside the reactor and run for 90 min at RT. The concentrations were expressed as pg/mL.

All plasma samples were analyzed centrally at the Translational Research Laboratory ARCO Foundation at S. Croce and Carle Teaching Hospital in Cuneo, Italy, and assessed in duplicate or triplicate. The average of each duplicate was evaluated at each point.

### 2.4. Statistical Analysis

Responses were assessed according to RECIST version 1.1 every 8 weeks. Clinical benefit (CB) was defined as the sum of all complete responses, partial responses and stable diseases lasting at least 6 months.

Progression free survival (PFS) was defined as the time elapsed between the start of nivolumab and progression of disease or death from any cause, whichever occurred first, or the date of last follow up for censored patients.

OS was defined as the time elapsed between the start of nivolumab and death from any cause or the date of the last follow up for censored patients.

PFS and OS were evaluated using Kaplan–Meyer method, and relative hazard ratio (HR) was analyzed with Cox model.

Differences in the median cytokine values or cell populations were analyzed using non-parametric Mann–Whitney U test; Wilcoxon signed-rank test for paired samples was used for continuous variables. Receiving operation characteristic (ROC) was employed to find the best threshold for our variables. We performed a multivariate analysis using Cox model to adjust HR with covariates.

Mann–Whitney U test and Wilcoxon signed-rank test were performed with GraphPad v.5 (GraphPad Software, Boston, Massachusetts USA). ROC analysis, and Kaplan–Meyer and Cox model were performed with SPSS V.24 (IBM SPSS Statistics for Windows, Version 24.0. Armonk, NY, USA).

In all tests, a *p* value equal to or below 0.05 was regarded as significant. Benjamini–Hochberg (B-H) procedure was applied to decrease false discovery rate at 25% [15]. If not clear, *p*-value was considered NS (not significant) for this percentage. The data that support the findings of this study are available from the corresponding author, [A.A.], upon request.

## 3. Results

### 3.1. Patient Population

A total of 22 patients were accrued between 19 March 2018 and 11 September 2020. Their median age was 67; their median ECOG PS was 0. HPV positivity was detected in three patients with an oropharynx tumor. The primary tumor site was detected in the larynx in 7 patients, and oral cavity, oropharynx and hypopharynx in 5 patients each. Main patients’ characteristics are reported in Table 1.

### 3.2. Treatment Effects

Overall, 6 patients (27%) achieved CB: 4 partial responses (18%) and 2 long-lasting disease stabilizations (9%).

The median PFS and OS of the whole population was 2.1 months (range 0.5–4.4) and 3.4 months (range 0.7–34.2), respectively (Appendix A).

We focused our analysis on OS. According to the grouping system described above, we divided our population as follows:

Group 1: sixteen patients with OS below or equal to 6 months (median OS 2.5 months; range 0.7–5.1).

Group 2: six patients with OS above 6 months (median OS 10.1 months; range 6.2–34.1).

We analyzed the distribution of the selected 18 cytokines (Appendix A).

Among them, ROC analysis identified those that could have a correlation with OS, matching group 1 against group 2.

IL-6, IL-8, IL-10 and TGF-β were able to differentiate group 1 from group 2 with good specificity (*p* < 0.2) (Appendix A).

TGF-β, IL-6 and IL-8 levels at T0 were higher in group 1 compared to group 2 (Appendix A), albeit only TGF-β met requisites for the false discovery rate (B-H).

In addition, ROC analysis also identified the threshold values for these cytokines, and was able to differentiate group 1 from group 2. ROC analysis: 336.85 pg/mL for TGF-β, 15.88 pg/mL for IL-6, 10.95 pg/mL for IL-8 and 3.25 pg/mL for IL-10. Using these values as cut-off points, we found a significant difference in the median OS for all these variables: values under the cut off correlated with a better OS. Kaplan–Meier graphs for these variables were plotted and are shown in Appendix A.

Moreover, we tested several covariates and we found that patients above the median age (67 years) had a better OS compared to patients below the median age, *p* = 0.031 (median OS 5.03 months, range 0.7–34.2; median OS 3.34 months, range 0.98–6.2, respectively).

Moreover, to evaluate the hazard ratio, a cox proportional regression analysis was performed. Patients with IL-6, IL-10 and TGF-β values under the threshold showed a longer OS: HR = 0.236 (95% C.I. 0.089–0.622), 0.262 (95% C.I. 0.102–0.672) and 0.170 (95% C.I. 0.046–0.623), respectively.

IL-8 and age subdivisions were not tested because they did not meet the proportional hazard.

### 3.3. Multivariate Cox Model

We realized a multivariate Cox model to test IL-6, IL-10 and TGF-β to stratify for age (Table 2). Age (above or below the median) was used for stratification and IL-6, IL-10 and TGF-β were used as covariates. Only patients with IL-6 and TGF-β values below the cut off showed a significantly lower HR, favoring better survival (Table 2).

### 3.4. Longitudinal Analysis

Circulating cytokines were analyzed by comparing the baseline values (T0) and their value after 3 courses of nivolumab (T1) to identify the modifications what were potentially induced with Nivolumab. However, due to the poor survival of our population, only 17 patients have been assessed at T1. At T1, the values of four cytokines (CCL-4, IL-15, IL-2 and CXCL-10) increased more than 50% compared to T0, even if they did not meet the requisites for the false discovery rate (B-H test) (Appendix A). Interestingly, all of them belong to the Th1 cytokines. However, the observed increase of these four cytokines is not associated with better survival since it was observed in both group 1 and group 2.

## 4. Discussion

Nivolumab induced a limited number of objective responses in our patient population.

This observation is in line with other clinical studies reporting modest activities of ICIs in patients with head and neck cancer previously treated for relapsed/metastatic disease [16].

Compared to Check-Mate (CM) 141, the study leading to nivolumab approval in this setting of patients, OS in our population is very poor (median OS = 3.4 months compared to 7.5 months in CM141) [17]. This result may be caused by the worse characteristics of the population accrued in our study. For instance, comparing to CM141, we enrolled 23% of patients with hypopharynx disease versus 3% in CM141. This subsite shows the poorest outcome among all HNCs [18]. Moreover, only 23% of our patients had their primary site of cancer located in the oropharynx, a subsite that shows a relatively good outcome [19], versus 50% in CM141. In our series, 3/22 patients had ECOG PS = 2, compared to 1/240 in CM141. ECOG PS is an important predictor of poor response to ICIs in recurrent/metastatic HNCs [20]. Moreover, females comprised 36% of the population compared to 18% in CM141, and it is known that females benefit less from immunotherapy [21]. Finally, current heavy smokers (>10 p/y) represent 64% of our population. This information is not available in the CM141 study; however, 79% of patients were current (either heavy or light) or past smokers. This data suggests that the number of current heavy smokers in CM141 may be lower than in our study, and it is known that heavy smokers benefit poorly from immunotherapy [22] due to the effects of smoking in gene expression and TME structure [23].

The poorer clinical characteristics of our patient population, compared to CM141, drove our decision to select an OS of 6 months as the cut-off point. Indeed, the median overall survival of the arm administered with nivolumab in the CM 141 study was 7.5 months (95% CI 5.5–9.1) [17]. Due to the poorer characteristics of our population, the cut off of 6 months, which is close to the lower limit but still within the limit of confidence of the median OS observed, was considered reasonable.

The 18 cytokines selected in our study are extensively included by other authors in panels aimed to identify prognostic/predictive biomarkers in other solid tumors [24,25]. Moreover, the same cytokines were tested in other cancers by our group, showing to be useful in identifying potential biomarkers [26,27].

In our study, the analysis of the selected variables at T0 allowed the identification of threshold values of four cytokines (IL-6, IL-10, IL-8 and TGF-β) and their capability of differentiating between patients with a good or poor OS.

Other authors have already identified a relationship between the levels of these cytokines and OS.

Chang et al. demonstrated that IL-6 is an independent marker of OS [28] in patients with oral squamous cell carcinoma. Hao et al. showed that IL-6 correlates with poor OS in squamous cell cancer of the larynx [29]. Some authors highlighted that the pretreatment level of IL-6 represents a good biomarker predicting OS in patients with HNSCC [30,31]. IL-6 plays a crucial role in immunoregulation and in tumor progression, and its overproduction is generally associated with dismal prognosis.

In our study, levels of IL-6 over the threshold value of 15.88 pg/mL were related to poor OS despite nivolumab treatment.

IL-10 is a pleiotropic immunosuppressive cytokine that can promote tumor metastasis [32]; many immune cells produce IL-10 [33,34]. Despite its recognized immunosuppressive role, the final effect of IL-10 is controversial [35,36].

However, several studies have shown a direct correlation between IL-10 and poor prognosis in melanoma, lung cancer, T/NK lymphoma and HNSCC [37,38,39]. In HNSCC, there is great interest in determining plasma levels of IL-10 as a potential biomarker of poor prognosis [40,41]. In our study we have observed that IL-10 levels above 3.25 pg/mL are associated with poor OS, notwithstanding nivolumab exposure.

TGF-β is a key factor in many physiological functions from the homeostasis of tissues to wound repair [42]. Notwithstanding it is considered a major immune suppressive factor, and TGF-β may have both oncogenic and tumor-suppressive effects [43,44]. The antitumor behavior of TGF-β could be explained by its cytostatic and proapoptotic effects [45]. However, the overexpression of TGF-β, mainly due to tumor cells [46], induces an immunosuppressive environment that promotes cancer progression [47]. Therefore, albeit not yet reported in patients with HNSCC treated with immunotherapy, it is not surprising that our patients with higher TGF-β levels (above the threshold of 336.85 pg/mL) exhibited a poor OS.

Schalper et al. [48] has pointed out that IL-8 could be a promising serum biomarker suitable for patients treated with ICIs, and elevated levels of this cytokine correlate with poor PFS and OS. In line with Schalper et al. data, IL-8 above the threshold of 10.95 pg/mL, is linked with a shorter OS in our patient’s population.

In our series, elderly patients (>67 years), had a better OS compared to younger patients. Bastholt et al. and Castro et al. have already reported that elderly patients treated with ICIs have a better OS compared to their younger counterparts [21,49].

Therefore, we performed a multivariate Cox analysis stratified for median age to put together all the three variables (IL-6, IL-10, TGF-β), which resulted in a proportional hazard with a significant impact on OS. IL-8 was excluded by Cox analysis because, as previously specified, the risk was not proportional over time.

Within the age strata, we observed that the only variables retaining a significant HR for better survival at T0 were circulating IL-6 and TGF-β. This might suggest that IL-6 and TGF- β baseline values below the threshold of 336.85 pg/mL and 15.88 pg/mL respectively, could be used to identify patients reaching longer survival with nivolumab treatment.

Longitudinal analysis T0-T1 shows that CCL-4, IL-15, IL-2 and CXCL-10 considerably increase between T0 to T1. Interestingly, all these cytokines have a Th1 effect. CCL4, among other effects, recruits dendritic cells from lymph nodes, a critical step to reactivate an immune response against a tumor [50]. IL-15 is a critical cytokine for NK cells development and expansion [51], IL-2 induces similar effects in CD8+ cells [52] and CXCL-10 is a chemokine responsible for recruitment of T cells, particularly in CD8+ cells, in the TME [50]. All these effects favor the modification of the TME toward an inflamed immune phenotype, which is associated with the best response to ICIs [2].

Surprisingly, the modulation of these 4 cytokines occurred in both groups 1 and 2 or, in other words, regardless of the survival of patients over or below 6 months, suggesting that immune suppression in many patients is stronger than the positive effect of nivolumab.

This study has some important limitations. First, the small number of patients prevents the possibility to use a subset of patients to model data to perform a machine learning approach with training and a test group. Therefore, the small sample size limits the generalizability of our findings, and the statistical values must be considered with caution. Second, we cannot exclude that long survivors mainly include patients with a high expression of PD-L1. In our study, PD-L1 expression was not requested in line with the EMA approval of nivolumab.

However, in our opinion, this exploratory study generates some interesting data. First, TGF-β is a known unfavorable prognostic marker in patients with HNSCC [53] that are treated with conventional therapies [54]. However, this study shows that a baseline level of circulating TGF-β over the identified threshold, is an unfavorable prognostic marker in patients with HNSCC that are treated with immunotherapy, which is an aspect not yet reported in the literature [55].

Second, the major changes observed between T0 and T1 are limited to Th1 cytokines. This finding may underline that the effect of nivolumab exceeds the PD-1/PD-L1 axis blockade, inducing positive pro-inflammatory modification in the TME. However, this change does not correlate with survival, suggesting that the positive effect of nivolumab used alone on Th1 cytokines may be not strong enough to counteract the immunosuppressive microenvironment of our patient population.

This finding supports the use of immune checkpoint inhibitors combined with other immune drugs directed, for example, against Th2 cytokines.

In conclusion, in our small population with poor clinical characteristics, our study highlights the negative prognostic role of IL-6 and TGF-β. Nivolumab treatment is associated with a positive modulation of some Th1 cytokines; however, at least in our patients, it does not correlate with a better outcome.

## Figures and Tables

**Table 1 cancers-15-05257-t001:** Patients’ baseline characteristics.

Characteristics	Number
Age (median, range)	67 (48–84)
Sex (M/F)	14/8
ECOG PS (median, range)	0 (0–2)
TNM at diagnosis	
T1–2/T3–4	11/11
N0–1/N2–3	15/7
M0	22 (100%)
Site of relapse	
Loco-regional	2
Distant metastasis	11
Both	9
Primary site	
Oral cavity	5 (22.7%)
Larynx	7 (31.8%)
Hypopharynx	5 (22.7%)
Oropharynx	5 (22.7%)
HPV positive	3 (60.0%)
Previous treatments	
Up-front treatments	22
Surgery	8 (36.3%)
Radiotherapy	2 (9.0%)
Concurrent chemo-radiotherapy	9 (40.9%)
Concurrent cetuximab-radiotherapy	3 (13.6%)
Previous treatment for recurrent/metastatic disease	22
None *	8 (36.3%)
One previous line (chemotherapy)	12 (54.5%)
Two or more previous lines (chemotherapy)	2 (9.0%)
Smoking	
Never, passed and current but </=10 p/y	8 (36.4%)
Current smokers, >10 p/y	14 (63.6%)

Legend: ECOG PS, Eastern Cooperative Oncology Group performance status; HPV, human papilloma virus; p/y, pack year; *, patients relapsed within 6 months from definitive platinum-based chemoradiotherapy.

**Table 2 cancers-15-05257-t002:** Multivariate Cox regression stratified for median age.

			95% C.I. for HR	
Variables	HR	*p* Value	Lower	Upper
IL-6				
>15.88 pg/mL	1			
≤15.88 pg/mL	0.168	0.028	0.035	0.828
TGF-β				
>336.85 pg/mL	1			
≤336.85 pg/mL	0.149	0.033	0.026	0.861
IL-10				
>3.25 pg/mL	1			
≤3.25 pg/mL	0.274	0.071	0.067	1.116

Legend: HR, hazard ratio; C.I., confidence level; IL, interleukin; TGF, transforming growth factor.

## Data Availability

Data availability will be provided by the ARCO lab (S. Croce e Carle, Cuneo, Italy) upon request to A.A.

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
