# Peer review of "Baseline Values of Circulating IL-6 and TGF-β Might Identify Patients with HNSCC Who Do Not Benefit from Nivolumab Treatment"

_cancers, 2023, doi:10.3390/cancers15215257_

Round 1

Reviewer 1 Report

Comments and Suggestions for Authors

The study provides valuable insights into the investigation of potential biomarkers for head and neck cancer patients undergoing nivolumab treatment. However, there are several shortcomings in this manuscript. Firstly, the study's small sample size of 22 patients limits the generalizability of its findings and raises concerns about statistical robustness. Additionally, the patient population appears to have poor clinical characteristics, which may not accurately represent the broader head and neck cancer population. Furthermore, the use of ROC analysis to identify potential biomarkers at T0 without validation in an independent cohort raises questions about the reliability of the identified markers.

The introduction provides a reasonable background for the study, highlighting the significance of immunotherapy in treating recurrent/metastatic head and neck squamous cell cancer and the challenges associated with identifying patients who will benefit from immune checkpoint inhibitors (ICIs). However, it lacks specific details about the scope and objectives of the study, making it challenging for readers to understand the study's precise research questions and hypotheses. Additionally, while it mentions the potential of cytokines as biomarkers, it does not clearly outline the rationale for selecting the specific cytokines under investigation in this study or how they might be relevant to the research objectives.

In the methodology section, the retrospective allocation of patients into two groups based on a six-month overall survival (OS) cutoff point may introduce bias, as this arbitrary cutoff might not accurately reflect the underlying patient characteristics or treatment response. The methodology does not address potential confounding variables or covariates that may influence the cytokine measurements. Moreover, the manuscript lacks a clear description of the treatment protocol, which is essential for understanding the context of the study. Additionally, the results are somewhat fragmented which make it challenging for readers to follow the logical flow of the analysis.

While the discussion delves into the differences in patient characteristics between this study and previous trials, it does not sufficiently explore the potential impact of these differences on the observed outcomes, leaving a gap in understanding why this patient population had a much worse overall survival. Secondly, the discussion lacks a critical evaluation of the limitations of the study, including the small sample size, the use of an arbitrary six-month OS cutoff, and the potential confounding factors that were not accounted for in the analysis. Furthermore, the discussion briefly mentions the modulation of Th1 cytokines in response to nivolumab treatment but does not provide a comprehensive interpretation of the clinical significance of these changes, leaving readers wondering about their relevance. 

Comments on the Quality of English Language

Minor editing of the English language is required

Author Response

Comments and suggestion for the Authors:

Point 1

The study provides valuable insights into the investigation of potential biomarkers for head and neck cancer patients undergoing nivolumab treatment. However, there are several shortcomings in this manuscript. Firstly, the study's small sample size of 22 patients limits the generalizability of its findings and raises concerns about statistical robustness.

Authors agree with the Reviewer; Indeed, they have already underlined this limit in the ‘conclusion’ section of the abstract and in final part of the discussion (‘The study has some important limitations.’). However, due to the importance of this point, as underlined by the Reviewer, we have further stressed this concept in the discussion section, adding the following statement: ‘Therefore, the small sample size limits the generalizability of our findings and the statistical value must be considered with caution’.  (See text, lines 291 – 293)

Point 2

Additionally, the patient population appears to have poor clinical characteristics, which may not accurately represent the broader head and neck cancer population.

Yes, this population has poor clinical characteristics. We have stressed this data in many parts of the manuscript. However, the Nivactor study accrued patients from the real-life (we have added information on the characteristics of the Nivactor study as suggested by the Reviewer in a following point). For this reason, albeit we confirm the poor clinical characteristics of our population, we can reassure the Reviewer that they remain in the range of typical clinical characteristics of the head and neck cancer population.

(See also: Ramakrishnan K et al: Real-world time on treatment with immuno-oncology therapy in

recurrent/metastatic head and neck squamous cell carcinoma. Future Oncol. (2021) 17(23), 3037–3050; Revesz M et al.: The characteristics of head and neck squamous cell cancer in young adults: A retrospective single-center study.  Pathology & Oncology Research (2023) DOI 10.3389/pore.2023.1611123)

Point 3

Furthermore, the use of ROC analysis to identify potential biomarkers at T0 without validation in an independent cohort raises questions about the reliability of the identified markers.

The Reviewer correctly observes that the small sample size of the population, does not allow us to give a definitive answer about the value of the identified biomarkers. An independent cohort of validation would be necessary. Unfortunately, 22 patients are not enough to split them in a training and test group. We had already introduced this message of caution in the original discussion section (‘The study has some important limitations. First, the small number of patients prevents the possibility to use a subset of patients to model data to perform a machine learning approach with training and a test group.’). However, to further stress this concept, we have added in the text the following statement: ‘Therefore, the small sample size limits the generalizability of our findings and the statistical value must be considered with caution.’ (See text, lines 291 – 293)

This study represents a starting point for other studies to test and validate our finding. In other words, this is an exploratory study, as already underlined in the original discussion section (‘However, in our opinion, this exploratory study generates some interesting data’).

(See text, line 296)

Point 4

The introduction provides a reasonable background for the study, highlighting the significance of immunotherapy in treating recurrent/metastatic head and neck squamous cell cancer and the challenges associated with identifying patients who will benefit from immune checkpoint inhibitors (ICIs). However, it lacks specific details about the scope and objectives of the study, making it challenging for readers to understand the study's precise research questions and hypotheses.

We modified introduction according the suggestion. Among other minor changes, we modified in particular the final part of the introduction as it follows, to clarify the research questions and hypothesis: ‘The aim of the present study was to test a panel of circulating cytokines at two time points, T0 (baseline) and T1 (during treatment), to identify possible biomarkers in head and neck cancer (HNC) patients treated with the ICI nivolumab. Furthermore, longitudinal changes possibly observed in the same cytokines, could add information on the effect of nivolumab.

(See text, lines 63 – 67)

Point 5

Additionally, while it mentions the potential of cytokines as biomarkers, it does not clearly outline the rationale for selecting the specific cytokines under investigation in this study or how they might be relevant to the research objectives.

We have already briefly explained the biological reasons that led us to identify this panel of cytokines in the materials and methods section. However, to go in deep in this matter according to the Reviewer suggestion, we have added this topic in the discussion section (‘The 18 cytokines selected in our study are extensively included by other Authors in panels aimed to identify prognostic/predictive biomarkers in other solid tumors (19A, 19B). Besides, the same cytokines have been tested in other cancers also by our group, showing to be useful to identify potential biomarkers [19C, 19D]).

(See discussion section, lines 229 - 232)

Point 6

In the methodology section, the retrospective allocation of patients into two groups based on a six-month overall survival (OS) cutoff point may introduce bias, as this arbitrary cutoff might not accurately reflect the underlying patient characteristics or treatment response.

We decided to select an OS of 6 months as a cut-off considering that in the Checkmate 141 study (ref 17) the median overall survival of the nivolumab arm was 7.5 months (95% CI 5.5 – 9.1). The characteristics of our population, as mentioned in the article, were poorer than the population accrued in Checkmate 141, therefore we have suggested a cut-off of 6 months which is close to the lower limit, but still within the limit of confidence of the median OS observed in Checkmate 141.

We have explained this point in the discussion session (The poorer clinical characteristics of our patient population compared to CM141, drove our decision to select an OS of 6 months as cut-off. Indeed, the median overall survival of the nivolumab arm in CM 141 study was 7.5 months (95% CI 5.5 – 9.1) (16). Due to the poorer characteristics of our population, the cut-off of 6 months which is close to the lower limit, but still within the limit of confidence of the median OS observed, was considered reasonable.).

(see text lines 223 - 228)

Point 7

The methodology does not address potential confounding variables or covariates that may influence the cytokine measurements.

We cannot consider more covariates in the multivariate Cox model regression, because usually 1 covariate for ten events is recommended. We have already exceeded in part this recommendation using three variables. Despite this assumption, in a univariate model our confounding variables are not significant and not proportional, so we considered in the multivariate analysis, only the three variables that were significant and proportional at the univariate.

See text, lines 181 - 182

Point 8

Moreover, the manuscript lacks a clear description of the treatment protocol, which is essential for understanding the context of the study.

We agree with the Reviewer. The lack of specific details about the Nivactor study in the manuscript is a serious shortcoming. We have added details on the study at the beginning of section 2 (Materials and methods): Briefly, the Nivactor study is a single-arm, open-label, multicenter, phase IIIb real-life clinical trial. The study aimed to verify in an unselected patient population with recurrent/metastatic HNC previously treated with chemotherapy, the toxicity of nivolumab based immunotherapy.

(See text, lines 72 -75)

Point 9

Additionally, the results are somewhat fragmented which make it challenging for readers to follow the logical flow of the analysis.

We edited many points in results section, following the Reviewer’s observation. We noted that the fragmentation was at least in part due to an inadequate English language, and so some changes have been introduced to improve the English used.

3.2 Treatment effects:

  • We added ‘range’ after the median values of PFS and OS (see text, lines 151 - 152 )
  • We added ‘values under the cut-off correlated with better OS.’ (see text, lines 171 - 172)
  • We added the range values after the median OS of patients older or younger of 67 (see text, lines 175 - 176)
  • We added ‘Besides, to evaluate hazard ratio, a cox proportional regression analysis was performed.’. This sentence is necessary to join two different analyzes conducted in subchapter 3.2. Its lack, was probably a major cause of the interruption in the logical flow of the analysis noted by the reviewer. (See text, lines 177 - 178).
  • All the remaining changes have bee made to improve English style. (See text, line 206/207; 220; 229)

Point 10

While the discussion delves into the differences in patient characteristics between this study and previous trials, it does not sufficiently explore the potential impact of these differences on the observed outcomes, leaving a gap in understanding why this patient population had a much worse overall survival.

We agree that the discussion section did not explain adequately the potential impact of the differences in patient characteristics between the present study and the CM141 study. We have added information and references to more clearly explain why these differences may justify the worse overall survival of our patients.

After editing, the discussion of this point is as it follows: ‘For instance, comparing to CM141, we enrolled 23% of patients with hypopharynx disease versus 3% in CM141. This subsite shows the poorest outcome among all HNC (18). Moreover, only 23% of our patients had their primary site in oropharynx, a subsite that shows relatively good outcome (19), versus 50% in CM141. In our series, 3/22 patients had ECOG PS = 2, compared to 1/240 in CM141. ECOG PS is an important predictor of poor response to ICIs in recurrent/metastatic HNC (20). Moreover, females were 36% compared to 18% in CM141. It is known that females benefit less from immunotherapy (21). Finally, current heavy smokers (>10 p/y) represent 64% of our population. This information is not available in CM141 study, but 79% of patients were current (either heavy or light) or past smokers. This data suggests that the number of current heavy smokers in CM141 may be lower than in our study. It is known that heavy smokers have poor benefit from immunotherapy (22) due to the effects of smoking in gene expression and TME structure (23).’

See text, lines 267 - 291

Point 11

Secondly, the discussion lacks a critical evaluation of the limitations of the study, including the small sample size, the use of an arbitrary six-month OS cutoff, and the potential confounding factors that were not accounted for in the analysis.

We have already addressed these points during the editing process (see points 3, point 6 and point 7).

Point 12

Furthermore, the discussion briefly mentions the modulation of Th1 cytokines in response to nivolumab treatment but does not provide a comprehensive interpretation of the clinical significance of these changes, leaving readers wondering about their relevance.

Again, this Reviewer’s observation is correct. We have modified the discussion to add the tools necessary to understand the relevance of these changes.

After editing, the subchapter is as it follows: ‘Longitudinal analysis T0-T1 shows that CCL-4, IL-15, IL-2 and CXCL-10, considerably increase between T0 to T1. Interestingly, all these cytokines have a Th1 effect. CCL4, among other effects, recruits dendritic cells from lymph nodes, a critical step to reactivate immune response against the tumor [50], IL-15 is a critical cytokine for NK cells development and expansion [51], IL-2 induces similar effects in CD8+ cells [52], CXCL-10 is a chemokine responsible for recruitment of T cells, in particular CD8+ cells, in the TME [50]. All these effects favor the modification of the TME toward an inflamed immune phenotype which is associated with the best response to ICIs (2). Surprisingly, the modulation of these 4 cytokines occurred in both group 1 and 2 or, in other words, regardless of survival over or below 6 months, suggesting that immune suppression in many patients is stronger than the positive effect of nivolumab. 

(See text, lines 501 - 511) 

Reviewer 2 Report

Comments and Suggestions for Authors

This is a relevant article in the field of oncological therapy for head and neck cancer.

It is fundamental to investigate how the tumor microenvironment and how the tumor evades the immune system, being therefore fundamental the study of biomarkers that previously identify that patients can benefit from a biological treatment as monoclonal antibody. Cytokines being biomarkers with many clinical advantages. Cytokines are biomarkers with many clinical advantages for the prognosis and prediction of positive results of nivolumab.

The materials and methods presented are correct and the results regarding the effects of treatment are interesting, although some results are to be expected as the little activity of immunotherapy in cases of smoking... which only confirms what can be predicted and predicted. Thus confirms the evidence.

Much remains to be done in this area and I encourage further research on the IL-6,8,10 ... and also about the antitumor TGF-β which is promising. Studying age differences between patients is an exceptional idea because the results vary.

In short, the results presented in the article cannot be considered as definitive, because there is a long way to go, but it can continue to deepen gaps that researchers know and consider appropriate to achieve more reliable results. However, the research is rigorous and contains scientific quality of this research.

Some modifications to be made are:

Table 2 should have a title at the top of it, as shown in table 1.

It is necessary to review the bibliography in its format to standardize it, although the content and citations of the mimas are correct. That is, indicate for example the DOI in all references and in blue to highlight them and standardize the way references are cited. And also, the numbers of references in the text must be all the same; between square brackets or parentheses.

I encourage further research in this área.

Author Response

We thank the reviewer for his kind words and encouragement to continue research in this field.

Specific comments:

Table 2 should have a title at the top of it…

  • We have edited Table 2. (See text, Table 2)

It is necessary to review the bibliography in its format…

  • We have edited the bibliography according to the reviewer’s suggestions (See text, references)

Round 2

Reviewer 1 Report

Comments and Suggestions for Authors

The authors have addressed all comments and suggestions. I am confident to recommend the manuscript for publication in its present form.